

# A Python framework for efficient use of pre-computed Green's functions in seismological and other physical forward and inverse source problems

Sebastian Heimann[1], Hannes Vasyura-Bathke[2, 4], Henriette Sudhaus[3], Marius Paul Isken[3, 1],
Marius Kriegerowski[3, 4], Andreas Steinberg[3], and Torsten Dahm[1,4]

[1]GFZ German Research Centre for Geosciences, Potsdam, Germany
[2]Division of Physical Sciences and Engineering, King Abdullah University of Science and Technology, Thuwal, Saudi Arabia
[3]Institute for Geoscience, University of Kiel, Germany
[4]Institute for Earth and Environmental Sciences, University of Potsdam, Germany

**Correspondence:** Sebastian Heimann (sebastian.heimann@gfz-potsdam.de)

**Abstract.**

The finite physical source problem is usually studied with the concept of volume and time integrals over Green's functions (GF), representing delta-impulse solutions to the governing partial differential field equations. In seismology, the use of realistic Earth models requires the calculation of numerical or synthetic GFs, as analytical solutions are rarely available.

The computation of such synthetic GFs is computationally and operationally demanding. As a consequence, on-the-fly re-calculation of synthetic GFs in each iteration of an optimisation is time-consuming and impractical. Therefore, pre-calculation and efficient storage of synthetic GFs on a dense grid of source to receiver combinations enables efficient look-up and utilisation of GFs in time critical scenarios. We present a Python-based framework and toolkit - *Pyrocko-GF*- that enables pre-calculation of synthetic GF stores, which are independent of their numerical calculation method and GF transfer function. The framework

integrates a suite of established numerical forward-modelling codes in seismology, and can incorporate new user-specified GF calculation methods. *Pyrocko-GF* defines an extensible GF storage format suitable for a wide range of GF types, handling especially elasticity- and wave propagation problems. The framework assists with visualisations, quality control and exchange of GF stores, which is supported through an online platform that provides many pre-calculated GF stores for local, regional and global studies. The *Pyrocko-GF* toolkit comes with a well-documented application programming interface (API) for the

Python programming language to efficiently facilitate forward modelling of geophysical processes, e. g. synthetic waveforms or static displacements for a wide range of source models.



# 1   Introduction

Green's functions (GFs) are used abundantly to represent force excitations of many different processes inside the Earth. A GF source representation is a mathematical concept to synthesise finite spatio-temporal sources. The GF is defined as the force delta-impulse solution of the system of partial differential equations of the field problem (see e. g. Tolstoy, 1973, chapter 7). In physics, GFs obey the causality condition that allows us to formulate the response at the receiver as a function of retarded time relative to the time of force excitation. For a finite duration of force excitation at the source, this leads to a time convolution integral representation of such a finite-duration source. Sources that have a finite spatial extent are realised by summing over weighted, spatially distributed point sources, leading to a volume integral representation of the source. Combining both integrals, an excitation source can be represented in space and time. The concept of GF source representation was introduced by George Green in 1828 (see e. g. Cannell and Lord, 1993), and since then it is used to describe electromagnetic, mechanical and thermal problems, including elasto-static deformation, wave propagation, wave scattering, and fluid and pressure diffusion.

With a combination of point forces we can approximate the wave fields and deformation for a variety of different processes. For instance, the wave field and the deformation of the well known double-couple moment tensor representation closely resembles a distribution of dislocations on a rupture plane (e. g. Dahm and Krüger, 2014). Such representations of finite-volume sources often include higher order moments and not only single forces, leading formally to elementary solutions, which are spatial and temporal derivatives of the original GFs. In practice, often the nomenclature is not distinguished between the elementary solutions and the point-force GF, and all is denoted as *Green's functions*.

Solving the non-linear inverse problem of determining the parameters of sources with finite duration and finite extent is a computational costly effort. It involves the repeated numerical calculation of a large number of GFs for combinations of force excitations in the context of an optimization algorithm. During the iterations of an optimization the parameters of the force excitations change, e. g. times, locations, orientations and strength. If done on-the-fly, the GF calculation may strongly dominate the overall computational costs of the optimization and even render it impractical.

The structured storage of GFs in *GF databases* (*GF stores* in the following) allows to query pre-calculated GFs for individual source-receiver configurations, rather than time-consuming re-calculation. The conceptional architecture of this approach is sketched in Figure 1. Pre-calculated GFs have been used since long time for operational routine seismological source inversion (Dahm and Krüger, 2014, Tab. 3). Often, these are in-house developed database solutions, or other structured storage, that are linked to specific forward modelling and inversion codes. Some recent developments provide open GF databases for seismological applications, like e. g. Instaseis (Van Driel et al., 2015) or Syngine (IRIS DMC, 2015). The benefit for users is significant, because the expensive calculation of standard GF databases, particularly for global seismic velocity models, has not to be done repeatedly. Also errors in the GF calculation can be avoided using such approved GF databases. However, these rigid database schemes are restricted to the modelling method that has been used to create them, and they are confined to specific moment tensor applications.

The aim of our work is to provide a stand-alone open-source toolbox for the calculation and storage of GFs, suited for an easy integration into individual routines. To achieve this, we made our framework independent of the GF type and the GF calculation

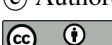


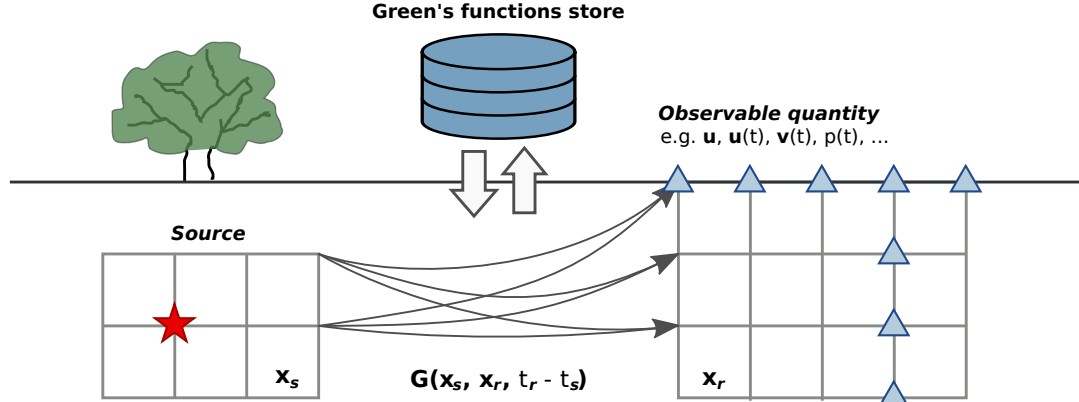

**Figure 1.** Concept of the Green's Function stores, the positions of the source $\boldsymbol{x}_s$ and the positions of receivers $\boldsymbol{x}_r$ form source-receiver pairs of Green's Functions $G(\boldsymbol{x}_s, \boldsymbol{x}_r, t_r - t_s)$ that are computed on a pre-defined grid for a specified time window. These GFs are saved in a store and can be utilised through the *Pyrocko-GF* Python API.

method that synthesises the physical quantities. The architecture of the GF stores is flexible to allow the storage of extra attributes, e. g. travel time tables. We put special efforts to find a good balance between stability, numerical performance and and user-friendly implementation. While being being part of the *Pyrocko* software library, our framework is easily linked to other seismological Python toolboxes, e. g. *Pyrocko* (Heimann et al., 2017) or *ObsPy* (Krischer et al., 2015). Similarly to the seis-

mological projects Instaseis and Syngine, we also openly share various existing GF stores on https://greens-mill.pyrocko.org.

The paper is organised as follows: section 2 describes the theoretical framework and GF definitions used in the proposed *Pyrocko-GF* infrastructure. Section 3 refers to the implementation design and section 4 gives examples of seismological applications.

## 2  Theoretical framework

### 2.1  Basic approach

For the following notations generally it applies that (1) vectors, tensors and matrices in symbolic notations are shown as bold-faced symbols, (2) index notations are expressed with the Einstein summation rule and (3) coordinates refer either, to Cartesian coordinate systems with $x$-$y$-$z$ or north-east-down ($n$-$e$-$d$), or to an axisymmetric polar coordinate system with

radial-tangential-vertical ($r$-$\varphi$-$z$, with $\varphi$ oriented clock-wise and $z$ oriented down). We denote a Green's function by $\mathbf{G} = \mathbf{G}(\boldsymbol{x}_r, \boldsymbol{x}_s, t_r - t_s)$, where $\boldsymbol{x}_r$ and $\boldsymbol{x}_s$ are receiver and source point positions, respectively. The time difference $t_r - t_s$ is the difference between the receiver time $t_r$ and the time $t_s$ of the delta-pulse force excitation at the source. We assume vector forces at the source and vector fields as observed variables at the receiver position. The GF is represented by a second-order tensor, as the three components of the observed vector can be excited by three forces acting in three different directions. Let $\boldsymbol{u}$

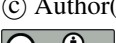



be the observed displacement vector field at the receiver located at the surface and, let $\boldsymbol{f}$ be the force vector density acting in the source volume $V$; then, the GF representation is given by

$$\boldsymbol{u}(\boldsymbol{x}_r,t_r) = \iiint\limits_{V} \mathrm{d}^3\boldsymbol{x}_s \int\limits_{-\infty}^{t_r} \mathrm{d}t_s \ \mathbf{G}(\boldsymbol{x}_r,\boldsymbol{x}_s,t_r-t_s) \, \boldsymbol{f}(\boldsymbol{x}_s,t_s) = \iiint\limits_{V} \{\mathbf{G}(\boldsymbol{x}_r,\boldsymbol{x}_s) * \boldsymbol{f}(\boldsymbol{x}_s)\} \ \mathrm{d}^3\boldsymbol{x}_s. \tag{1}$$

The observed quantity is a superposition of all forces acting in the source volume, convolved with the individual timing of
force excitation. In the second GF representation (Eq. 1) the convolution integral is replaced by the symbol $*$, omitting the time variable $t$. The convolution in Eq. (1) can be rewritten as a matrix equation in any coordinate system convenient for numerical implementation, with three-dimensional data vector $\boldsymbol{u}$ and source vector $\boldsymbol{f}$, and a 3x3 matrix $\mathbf{C}$ of GF components as

$$\mathbf{G}(\boldsymbol{x}_r,\boldsymbol{x}_s) * \boldsymbol{f}(\boldsymbol{x}_s) = \begin{pmatrix} C_{11} & C_{12} & C_{13} \\ C_{21} & C_{22} & C_{23} \\ C_{31} & C_{32} & C_{33} \end{pmatrix} * \begin{pmatrix} f_1 \\ f_2 \\ f_3 \end{pmatrix}. \tag{2}$$

While Eq. (1) is valid in general, a buried seismic source can also be represented by generalised force couples $\mathbf{m}$, localised
on a planar surface with an area $A$ (rupture plane). The generalised force couples $\mathbf{m}$ form a symmetric, second-order moment tensor density. In such a moment tensor representation, the GF tensor has 27 instead of nine components. It relates to the first and second-order spatial derivatives of the single-force GF (denoted by a comma-separated last index), as (e. g. Aki and Richards, 2002)

$$u_i(\boldsymbol{x}_r,t_r) = \iint\limits_{A} \{G^i_{j,k}(\boldsymbol{x}_r,\boldsymbol{x}_s) * m_{jk}(\boldsymbol{x}_s)\} \ \mathrm{d}A \qquad \boldsymbol{x}_s \in A, \tag{3}$$

with $i,j,k \in \{x,y,z\}$ in Cartesian coordinates. In $G^i_{j,k}$, the superscript index $i$ is the displacement direction, while the subscripts $j$ & $k$ indicate the force direction and force arm, respectively.

A matrix formulation of the index-notated Eq. 3 is obtained by mapping the symmetric moment tensor $\mathbf{m}$ into a source vector of length six. Accordingly, the Green's tensor becomes a 3 x 6 matrix:

$$\{G^i_{j,k}(\boldsymbol{x}_r,\boldsymbol{x}_s) * m_{jk}(\boldsymbol{x}_s)\} = \begin{pmatrix} C_{11} & C_{12} & C_{13} & C_{14} & C_{15} & C_{16} \\ C_{21} & C_{22} & C_{23} & C_{24} & C_{25} & C_{26} \\ C_{31} & C_{32} & C_{33} & C_{34} & C_{35} & C_{36} \end{pmatrix} * \begin{pmatrix} m_{11} \\ m_{22} \\ m_{33} \\ m_{12} \\ m_{13} \\ m_{23} \end{pmatrix}. \tag{4}$$

Due to the symmetry of the moment tensor, the GF matrix in general has only 18 independent components: $C_{i4} = G^i_{1,2}+G^i_{2,1}$, $C_{i5} = G^i_{1,3}+G^i_{3,1}$, and $C_{i6} = G^i_{2,3}+G^i_{3,2}$.




The given matrix formulation is convenient for an efficient numerical implementation in the GF store and it allows for a generalisation of mixed problems. For example, let $u$ in Eq. (3) be a mixed-data vector $b$ observed at the receiver, consisting of ground displacements $u$ and rotational motions $\omega$. Also, let the source vector $s$ be represented by a mixed-type force system, where force couples and single forces act together within the source volume $V$. Then, the matrix representation of the mixed

problem can be written as

$$b(t_r) \quad = \quad \iiint\limits_V \{\mathbf{C} * s\} \, \mathrm{d}^3 x_s \tag{5}$$

with

$$b \quad = \quad (u_n, u_e, u_d, \omega_n, \omega_e, \omega_z)^{\mathrm{T}}, \quad s = (m_{nn}, m_{ee}, m_{dd}, m_{ne}, m_{nd}, m_{ed}, f_n, f_e, f_d)^{\mathrm{T}} \quad \text{and} \quad \mathbf{C} = \left( \begin{array}{cc} \mathbf{C}^{\mathrm{MT}} & \mathbf{C}^{\mathrm{SF}} \end{array} \right).$$

Note that in Eq. (5) $\mathbf{C}$ contains GF components of different units. If the GF components $\mathbf{C}$ are known over the range of

potential source and receiver positions as well as over the relevant time lags, the observables due to arbitrary source models can be synthesised using Eq. (5), by approximating the temporal and spatial integrals with sums (e. g. Dahm and Krüger, 2014).

## 2.2 Green's tensor and source models in layered elastic media

The number of independent GF components can be further reduced for one-dimensional layered media of cylindrical symmetry in a flat-Earth model or, of spherical symmetry in a spherical-Earth model. For a GF tensor that is calculated for a station at

zero azimuth we use the variable $\mathbf{^0 G}$. Considering geometric symmetries at zero azimuth (north direction), P-wave motion and vertically polarised S-wave (SV) motion can only be excited by the moment tensor components $m_{xx}$, $m_{yy}$, $m_{zz}$, $m_{xz}$, and $m_{zx}$. Similarly, horizontally polarised S-wave (SH) motion can only be excited by the moment tensor components $m_{xy}$ and $m_{yx}$. This reduces $\mathbf{^0 G}$ to the form (Dahm, 1996; Heimann, 2011; Dahm and Krüger, 2014):

$$\mathbf{^0 G^x} = \left( \begin{array}{ccc} ^0G^x_{x,x} & 0 & ^0G^x_{x,z} \\ 0 & ^0G^x_{y,y} & 0 \\ ^0G^x_{z,x} & 0 & ^0G^x_{z,z} \end{array} \right), \mathbf{^0 G^y} = \left( \begin{array}{ccc} 0 & ^0G^y_{x,y} & 0 \\ ^0G^y_{y,x} & 0 & ^0G^y_{y,z} \\ 0 & ^0G^y_{z,y} & 0 \end{array} \right), \mathbf{^0 G^z} = \left( \begin{array}{ccc} ^0G^z_{x,x} & 0 & ^0G^z_{x,z} \\ 0 & ^0G^z_{y,y} & 0 \\ ^0G^z_{z,x} & 0 & ^0G^z_{z,z} \end{array} \right). \tag{6}$$

This reduces the number of independent GF components to ten (e. g. Mueller, 1985). These GF components depend only on the source depth, the distance to the receiver and the receiver depth.

A layered medium is invariant to rotation around $z$ at the source element. We exploit this symmetry to derive GFs in $r$-$\varphi$-$z$ direction at a station with a given azimuth, through a tensor rotation with the azimuth angle $\varphi$:

$$G^{i'}_{p,q} = R_{pj}(\varphi) \, R_{qk}(\varphi) \, ^0G^i_{j,k} \quad \text{or} \quad \mathbf{G^{i'}} = \mathbf{R}(\varphi) \, \mathbf{^0 G^i} \, \mathbf{R^T}(\varphi) \quad \text{with} \quad \mathbf{R}(\varphi) = \left( \begin{array}{ccc} \cos\varphi & -\sin\varphi & 0 \\ \sin\varphi & \cos\varphi & 0 \\ 0 & 0 & 1 \end{array} \right). \tag{7}$$

Note that index $i'$ is in the $r$-$\varphi$-$z$ coordinate system, while $i$ is in the local $x$-$y$-$z$ coordinate system at the source (equivalent to north-east-down). Thus, with Eq. (7), the convolution of $\mathbf{m}$ with $\mathbf{G}$ (Eq. 3) is derived component-wise as



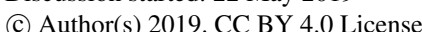


$$u_r = \left[G^r_{j,k} * m_{jk}\right]_r = g^r_1 * \left(m_{xx}\cos^2\varphi + m_{yy}\sin^2\varphi + m_{xy}\sin 2\varphi\right) \quad + g^r_2 * \left(m_{xz}\cos\varphi + m_{yz}\sin\varphi\right)$$
$$+ g^r_3 * m_{zz} \quad + g^r_4 * \left(m_{xx}\sin^2\varphi + m_{yy}\cos^2\varphi - m_{xy}\sin 2\varphi\right) ,$$
$$u_z = \left[G^z_{j,k} * m_{jk}\right]_z = g^z_1 * \left(m_{xx}\cos^2\varphi + m_{yy}\sin^2\varphi + m_{xy}\sin 2\varphi\right) \quad + g^z_2 * \left(m_{xz}\cos\varphi + m_{yz}\sin\varphi\right)$$
$$+ g^z_3 * m_{zz} \quad + g^z_4 * \left(m_{xx}\sin^2\varphi + m_{yy}\cos^2\varphi - m_{xy}\sin 2\varphi\right) ,$$
$$u_\varphi = \left[G^\varphi_{j,k} * m_{jk}\right]_\varphi = g^\varphi_1 * \left(\frac{1}{2}\left(m_{yy} - m_{xx}\right)\sin 2\varphi + m_{xy}\cos 2\varphi\right) \quad + g^\varphi_2 * \left(m_{yz}\cos\varphi - m_{xz}\sin\varphi\right) , \tag{8}$$

with

$$g^r_1 = {}^0 G^r_{x,x} , \qquad g^r_2 = {}^0 G^r_{x,z} + {}^0 G^r_{z,x} , \qquad g^r_3 = {}^0 G^r_{z,z}, \qquad g^r_4 = {}^0 G^r_{y,y} ,$$
$$g^z_1 = {}^0 G^z_{x,x} , \qquad g^z_2 = {}^0 G^z_{x,z} + {}^0 G^z_{z,x} , \qquad g^z_3 = {}^0 G^z_{z,z} , \qquad g^z_4 = {}^0 G^z_{y,y}$$
$$g^\varphi_1 = {}^0 G^\varphi_{x,y} + {}^0 G^\varphi_{y,x} , \qquad g^\varphi_2 = {}^0 G^\varphi_{y,z} + {}^0 G^\varphi_{z,y} .$$

The calculation of synthetic displacements is a linear combination of ten GF components $g^i_j$, here elementary seismograms, with the moment tensor components as weighting factors. These GF components $g^i_j$ are placed into a GF store. The GF components $g^r_4$ and $g^z_4$ are near-field terms and do not add to regional or teleseismic observations of dynamic displacements (Aki and Richards, 2002).

The GF representation from Eq. (8) can be re-ordered in the form of the matrix notation (Eq. 5)

$$\mathbf{b} = \begin{pmatrix} u_r & u_\varphi & u_z \end{pmatrix}^T , \quad \mathbf{s} = \begin{pmatrix} m_{xx} & m_{yy} & m_{zz} & m_{xy} & m_{xz} & m_{yz} \end{pmatrix}^T$$

$$\mathbf{C} = \begin{pmatrix} \left[g^r_1\cos^2\varphi + g^r_4\sin^2\varphi\right] & \left[g^r_1\sin^2\varphi + g^r_4\cos^2\varphi\right] & g^r_3 & \left[g^r_1 - g^r_4\right]\sin 2\varphi & g^r_2\cos\varphi & g^r_2\sin\varphi \\ -g^\varphi_1\frac{\sin 2\varphi}{2} & g^\varphi_1\frac{\sin 2\varphi}{2} & 0 & g^\varphi_1\cos 2\varphi & -g^\varphi_2\sin\varphi & g^\varphi_2\cos\varphi \\ \left[g^z_1\cos^2\varphi + g^z_4\sin^2\varphi\right] & \left[g^z_1\sin^2\varphi + g^z_4\cos^2\varphi\right] & g^z_3 & \left[g^z_1 - g^z_4\right]\sin 2\varphi & g^z_2\cos\varphi & g^z_2\sin\varphi \end{pmatrix} ,$$

where $\mathbf{b}$ is the observed data vector (here components of displacements), $\mathbf{s}$ is the source vector (here components of moment tensors), and $\mathbf{C}$ is the matrix form of the GF components. This form is suited for a mixed-type implementation (Eq. 5) and it also allows for a simple projection of $\{\mathbf{C} * \boldsymbol{s}\}$ *a priori* to components of the data vector. These projections can be useful for example, to project to the ray system (L-Q-T), to inclined borehole sensors, or to transform the $r$-$\varphi$-$z$ coordinate system to a $n$-$e$-$d$ coordinate system of the sensor.

## 3 Implementation and design

### 3.1 Green's function store architecture, schemes and storage format

Green's function stores consist of a discrete, sparse and truncated representation of the $g^i_j$ in Eq. (8). Time-dependent 1D arrays (GF traces) are stored for a finite grid of source and receiver positions, or of relative coordinates, resembling a finite spatial




extent (Fig. 1). The GF traces are stored in a single binary file `traces`, which allows for efficient reading of the stored data. Lookup indices for accessing the GF traces are stored in the binary file `index` (for technical details, see Heimann, 2019). Start and end times of the GF traces may be different for every node to allow compact storage of application-relevant portions of the waveforms (e. g. P or S phases, surface waves, etc.). A repeating end-point condition is assumed when extracting waveforms

from the store. This way, a step function can be represented by just two samples at every node (e. g. for the modelling of static displacement changes). The strategy we follow to build synthetic data vectors for arbitrary source-receiver locations by using the discretised GF is described below in section 3.2.

We categorise several store configuration types that are tailored to specific groups of source-medium-receiver configurations. The physics of the modelled process determines the number of GF components that are needed to represent the effects of a

source at a receiver site (Tab. 1). For example, in a planar layered medium at short distances, the simulation of a full moment tensor requires 10 GF components, a single force or isotropic source only needs 5 or 2 components, respectively. Also, for receivers located in the far-field, the number of neccessary components reduces. The combination of these categories defines the structure of the store (Tab. 1). Such a flexible design also allows to store GFs of other transfer functions, e.g. to simulate the poroelastic behaviour of a medium. This categorisation allows to provide efficient calculation, storage and reading of GFs

for problems of lower dimension. Consequently, the generation of synthetic quantitites, e.g. seismograms, is efficient.

**Table 1.** Green's tensor component schemes and configuration types for GF stores. Configuration types A and B stand for rotation-symmetric media with (A) common-depth receivers or (B) variable-depth receivers. Configuration type (C) is used for heterogeneous media with sources in a 3D volume (box) and receivers at fixed positions.

| Component scheme | Configuration type | Source type | Receiver distances |
|---|---|---|---|
| `elastic10` | A, B | moment tensor | all |
| `elastic8` | A, B | moment tensor | far-field only |
| `elastic5` | A, B | single force | all |
| `elastic2` | A, B | isotropic source | all |
| `poroelastic10` | A, B | flow rate, pressure | near field (quasi-static) |
| `elastic18` | C | moment tensor | all |

The range and spacing of source locations, depths and lateral source-receiver distances, along with the time span and the sampling rate of the GF traces and other meta data are defined in a configuration file, `config`. The `config` is defined in the human-readable structured YAML-format. This eases the inspection of the meta data and the file can easily be edited with any text editor. The configuration file can be created and edited either manually or automatically by using the API. For details on

the binary encoding and the YAML data structure, please see Heimann (2019).

In addition to the mentioned files `traces`, `index` and `config`, the store comprises two directories, `extra` and `phases`. They contain the individual GF method-related configuration file(s) and, in case of waveform problems, the travel time tables for user-specified arrivals. Providing travel time tables consistent with the GFs is useful in practical applications, e.g. to extract specific seismic phase arrivals.





Once created, the grid of traces in a GF store may be decimated in space and time. Such a downsampled version may be placed in an individual store or, in the sub-directory `decimated` of the parent GF store. A downsampled store can be read faster and synthetic quantities are calculated faster. This can be useful, e.g. to simultaneously simulate high frequency body waves and long period surface waves by using the full resolution store and the downsampled store, respectively.

## 3.2 Green's function computational backend modules

The Green's functions depend on the Earth model and the problem geometry. As scientific applications differ, several specialised numerical methods have been developed to calculate GFs. For instance, reflectivity-type wavenumber integration methods (e. g. QSEIS, Wang, 1999) are commonly used to calculate high-frequency body waves and surface waves from

teleseismic to local distances. Direct integration (e. g. GEMINI, Friederich and Dalkolmo, 1995) or hybrid methods (e. g. QSSP, Wang et al., 2017) are established for long-period global seismology on a spherical Earth. QSSP can also be used to simulate the coupling between solid earth, ocean and atmosphere, including gravity waves and infrasound, or to calculate the elasto-gravitational effect from elastic wave propagation. The program INSTASEIS (Van Driel et al., 2015) is suited to calculate approximative wave solutions for a 3D Earth structure with radial symmetry. Layered elasto-static and gravity problems

in seismology and volcanology are often approached with methods adopted from Haskell-type integration (e. g. PSGRN/P-SCMP, Wang et al., 2006). Poroelastic quasistatic problems of fully coupled deformation and flow in layered media can be handled using orthonormal Haskell-type integration (e. g. POEL, Wang and Kümpel, 2003).

Different GF computation methods are implemented with different model parameter conventions (e. g. for the coordinate systems, source model descriptions, etc.). They are also written in different programming languages (e. g. Fortran or C). There-

fore, we designed the structure of the GF store to be independent from the calculation method of the GFs. A standardised set-up and configuration is ensured by so-called *backends*, written as Python modules. These backends transfer the output of the respective GF computation method to the data-format of the GF store. Such a concept enables the implementation of various existing or future GF calculation methods. The structured filling process of the store allows parallel computation of GFs to efficiently exploit the computer systems at hand.

At the time of publication we have implemented backends for QSEIS, QSSP, PSGRN/PSCMP and POEL to be installed individually and to be used with the Python toolbox *Pyrocko* (Fig. 2a, and Heimann et al., 2017). In addition to these external code backends, *Pyrocko* includes its own method AHFULLGREEN. This backend calculates GFs for an analytical, homogeneous full space (Aki and Richards, 2002) including near-, intermediate- and far-field wave components. It can be applied to simulate acoustic emissions in mines or laboratory probes, or it can be useful for lecturing purposes. The chosen structure is not limited

to GFs of force excitation. Also, GFs simulating other processes could be handled through GF stores, e. g. flow rate and pressure pulses. Only the implementation of a method-related backend is needed here.





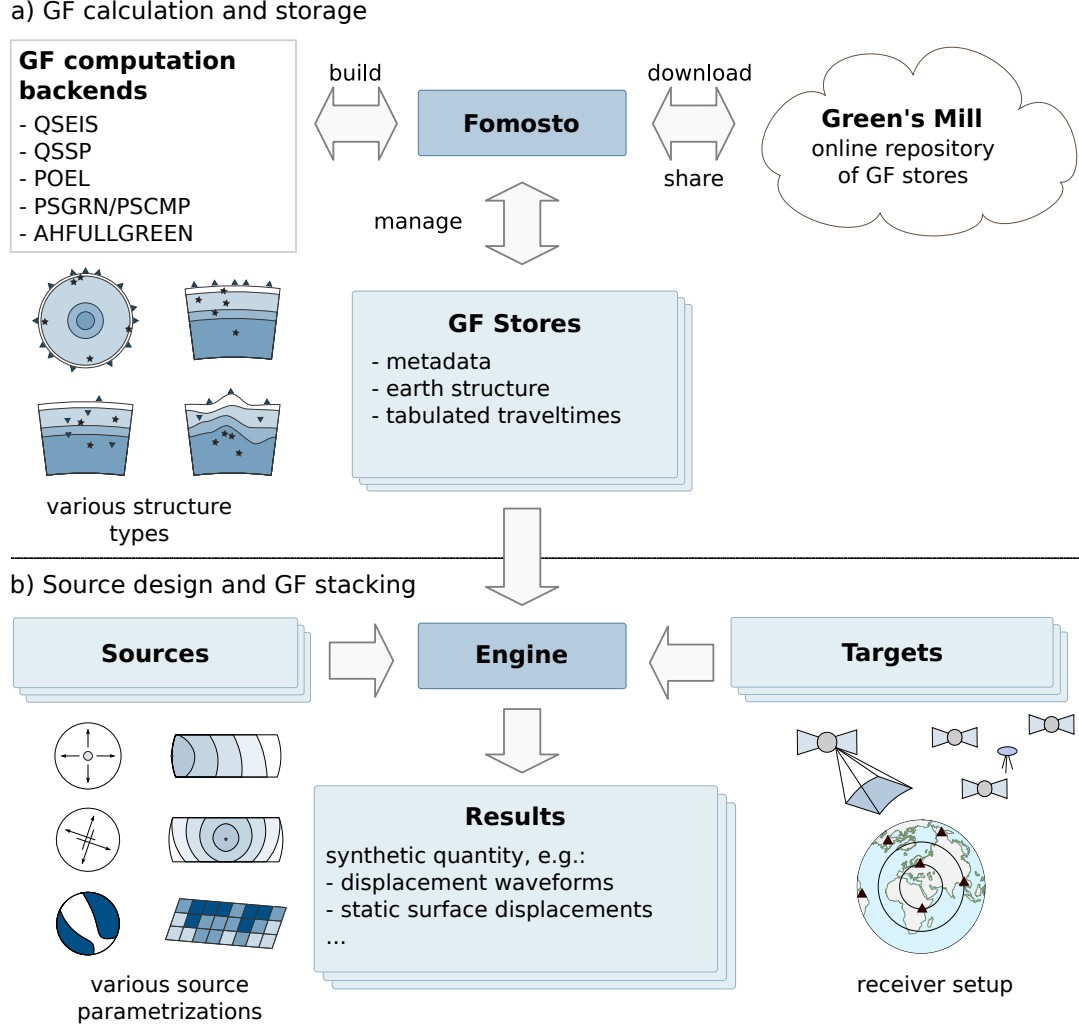

**Figure 2.** The architecture of *Pyrocko-GF* and object oriented implementation scheme. a) The *fomosto* program is a command-line interface (CLI) to create GFs using the backends, to inspect and manage GF stores. b) The *engine* is the central object that accesses the computed Green's Function stores and calculates the synthetic waveforms based on the specified source (e. g. moment tensor, double-couple source or finite-rectangular) and targets (e. g. virtual seismometers, GNSS stations, or InSAR line-of-sight displacements).

### 3.3 Source design

To simulate the physical quantity related to points of force excitations (sources), several GF traces from the pre-calculated store are combined. In the following, we illustrate the concept of source design based on a rectangular dislocation source that is finite in space and time, with uniform slip, a rupture nucleation point and a rupture velocity.

5    The calculation of an observed quantity of interest (e. g. seismic waveforms) for a point source with delta-force excitation at a particular source-receiver constellation is given in section 2. The observed quantity at the receiver is a linearly weighted





combination of the spatially closest GF components. Often, this requires interpolation to match the requested source-receiver configuration. The interpolation between neighbouring grid nodes can be simple nearest-neighbour or multi-linear interpolations. As described above, the specific combination of GF components is defined by the source type and the observed type of quantity (e. g. a full moment tensor or a single force, generating far-field waveforms or surface displacements, see Tab. 1).

To constrain the finite-duration of moment release at a source point, we extend the delta-force excitation to force excitations in time, which is known as the source time function (STF). Common STF types are triangular, half-sinusoidal, boxcar or smooth-ramp (Brüstle and Müller, 1983; Udias et al., 2014). All of these are defined through a source duration parameter and STF specific parameters, e. g. the peak-moment for a triangular STF. The temporal discretisation of finite-duration source models is corresponding to the temporal sampling of the stored GF traces, $\Delta t_{\mathrm{GF}}$.

The linear combination of several of these point sources allows to construct spatially finite sources. For example, a rectangular source can be expressed with point sources that are aligned on a rectangular grid outlining the area of the finite source. The source area is defined by the length and width of the rectangular plane, and it's strike and dip. In combination with the rake angle and the amount of slip on the rupture plane, the mechanism of the dislocation and the moment are defined. The nucleation point of the rupture across the fault is defined relative to the center of the rectangular plane.

The spatial distance of point sources on the plane is controlled by the spatial discretisation of the stored GFs ($\Delta z_{\mathrm{GF}}$ and $\Delta x_{\mathrm{GF}}$). The point source spacing is also constrained by the temporal sampling intervals $\Delta t_{\mathrm{GF}}$, and the minimum possible rupture velocity $v_{\mathrm{rupt,min}}$. The distance between point sources, $\Delta_{\mathrm{src}}$, should either be smaller than half the minimum GF grid spacings or half the distance it takes the rupture to propagate in $\Delta t_{\mathrm{GF}}$:

$$\Delta_{\mathrm{src}} < \frac{1}{2}\min\left(\Delta z_{\mathrm{GF}}, \Delta x_{\mathrm{GF}}, \Delta t_{\mathrm{GF}} \cdot v_{\mathrm{rupt,min}}\right). \tag{9}$$

Accordingly, the integer number of equally-spaced point sources $N_{\mathrm{L,src}}$ along the length of a rectangular, planar source $L_{\mathrm{src}}$ is:

$$N_{\mathrm{L,src}} = 1 + 2 \cdot \left\lceil \frac{L_{\mathrm{src}}}{\min(\Delta z_{\mathrm{GF}}, \Delta x_{\mathrm{GF}}, \Delta t_{\mathrm{GF}} \cdot v_{\mathrm{rupt,min}})} \right\rceil. \tag{10}$$

The same relations apply for the number of sources across the rectangular plane $N_{\mathrm{W,src}}$ of width $W_{\mathrm{src}}$. Thus, the total number of point sources $N_{\mathrm{src}}$ that build the finite source is $N_{\mathrm{src}} = N_{\mathrm{L,src}} \cdot N_{\mathrm{W,src}}$. In case the point sources contribute equally to the

moment of the finite source, the point source moment is the total moment divided by the number of point sources $N_{\mathrm{src}}$. In general, the sum of all point source moments is the total moment of the discretised source.

    Finally, we define the rupture propagation across the rupture plane by the rupture velocity $v_{\mathrm{rupt}}$. The rupture starts at the nucleation point and from there propagates radially across the fault. The other point sources start rupturing with a time shift of $t_{\mathrm{shift},i}$ at the $i$th point source. This time shift depends on the distance $d_i$ between the point source and the nucleation point and

the mean rupture velocity $\overline{v}_{\mathrm{rupt},i}$ between these two points:

$$t_{\mathrm{shift},i} = \frac{d_i}{\overline{v}_{\mathrm{rupt},i}}. \tag{11}$$

With a similar strategy many more finite-source types can be realised using *Pyrocko-GF* (for details see section 3.5 and the online documentation at https://pyrocko.org/docs).



Once the source model has been discretised into an ensemble of point sources in space and time, a discrete approximation of (1) can be used to calculate a synthetic seismogram or any other observable quantity at a receiver site. The sample $u[i]$ at discrete time $i$ can be computed from a sum of weighted and time-shifted GF contributions as

$$u[i] = \sum_j G[j, i - k[j]]w[j] \qquad (12)$$

where $w[j]$ are weighting coefficients and $k[j]$ are delay times. The index $j$ selects the stored GF trace and is a mapping of source position, receiver position, and the GF component index. If it is required to interpolate between the nodes of stored GFs, multiple such contributions from a neighbourhood in GF space are added. Similarly, time interpolation can be implemented by adding weighted and time shifted GF contributions. The weights $w[j]$ are computed considering (1) a rotation at the source (from the source coordinate system to the GF coordinate system), (2) a rotation at the receiver (from the GF coordinate system to the receiver component orientations) and (3) the involved interpolations. Our implementations of source design and GF

stacking approach in *Pyrocko-GF* take into account additional practical considerations to provide an efficient, flexible and easy-to-use modelling engine.

### 3.4   Computation and quality checks of Green's function stores

With the user interface *fomosto* (FOrward MOdeling and Storage TOol) we provide a tool to create, manage and inspect

GF stores. *fomosto* builds the GF stores and facilitates the calculation of the GFs through the backends. Furthermore, *fomosto* features chopping of the GF traces to travel-time-relative time windows before their insertion into the GF store. This reduces the amount of stored data and removes synthetic seismogram parts prone to high contamination by numerical noise. Another store-managing feature of *fomosto* is the spatial and temporal decimation of the GF store (Sec. 3.1). Last but not least, *fomosto*-built GF stores can be accessed by the *Pyrocko* project for many forward-modelling applications (see following Sec. 3.5).

The quality control of the ensemble of GF traces is not trivial. GF stores can contain GFs for a large number of source and receiver positions such that a complete visual control by the scientist is impossible. In order to facilitate quality control, *fomosto* offers several options for visual inspection. E. g. in a report generated with its *report* sub-command, the GF traces are assembled in specified time windows, in pre-defined frequency ranges together with predicted travel time arrivals (Fig. 3). The standard report produces plots of displacement and velocity traces, plots of maximum amplitudes and plots of displacement

spectra from the GF traces in the analysed store(s). These plots help identifying spurious signals or numerical artefacts in synthetic seismograms. The reports can be customised through a dedicated configuration file. *fomosto* can also be used to visualise the meta data of the GF store, e. g. the Earth model of seismic velocities or the associated travel time tables.

Pre-calculated GF stores are provided for download on the *Green's Mill* web-service under https://greens-mill.pyrocko.org. As of today, GF stores for global and regional applications are provided for a set of different global-1D Earth models and

regional models adapted to the CRUST 2.0 crustal velocity models (https://igppweb.ucsd.edu/~gabi/crust2.html). GF stores developed during new applications and external projects can be uploaded to the GF store web page.





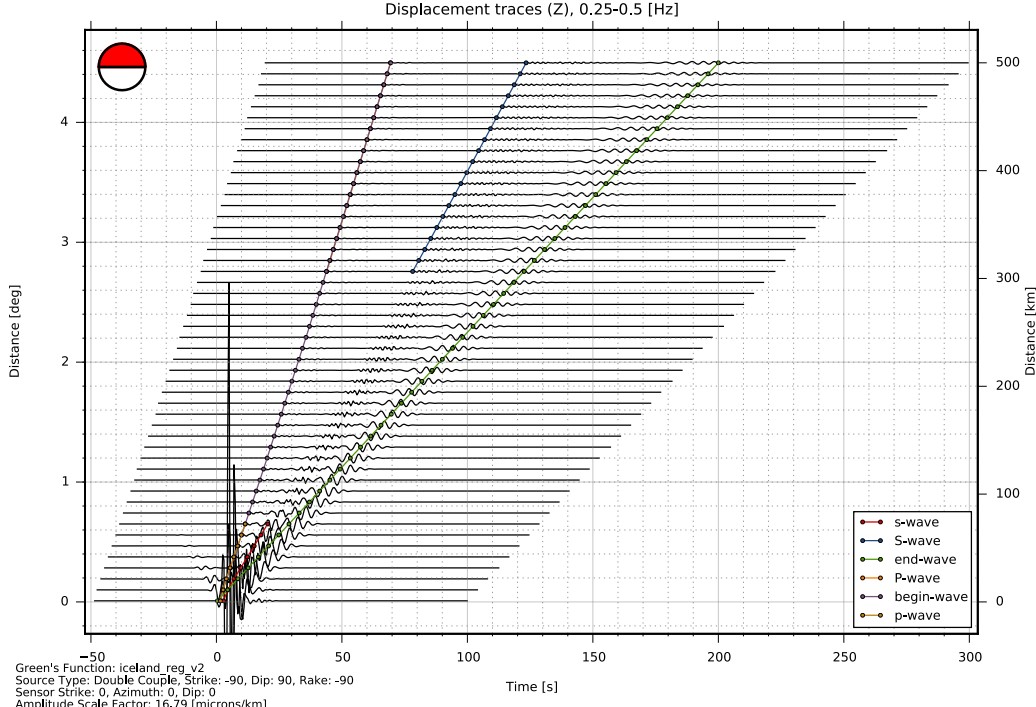

**Figure 3.** Synthetic seismogram distance section generated by the `report` sub-command of *fomosto* (generated by command `fomosto report single min_config` where *min_config* is a configuration file). Vertical component seismograms are calculated for a given moment tensor source for a given frequency range and distance range provided in GF store (0.25-0.5 Hz at 0-500 km distance). Theoretical first arrivals (P body waves) and other travel time curves are plotted together with synthetic seismograms. Such a display is useful to identify numerical artifacts. Here, an acausal numerical phase running with a negative velocity into negative times at distances below 50 km can be found.

### 3.5   Forward modelling programming interface

The *Pyrocko-GF* framework implements the described methods to utilise the GF stores for forward modelling of seismic wave-forms and near-field displacements. The framework is focused on various forward modelling applications in seismology and geophysics. The object-oriented programming model provides `Source` objects defining the dislocation source(s), `Target`
5   objects defining the modelling target (e. g. seismometer components or GNSS station), and the `Engine` object being respon-sible for forward modelling (Fig. 2b). The `Source` objects define the source properties, which may include the location, mechanism and origin time. The `Source` object is responsible for the discretisation of point- and finite-extent sources into their moment tensor representations for weighting of the store's GF traces (see Eq. 8). The `Target` object defines the parame-ters of the synthesised quantities such as location of receivers and the requested quantity type. Yet, seismic targets can provide
10   derivatives of the synthetic displacements (i. e. velocity and acceleration). The `Engine` connects the `Source` and `Target`



objects to the GF store. Based on the configuration of the `Source` and `Target`, the `Engine` extracts the required GF traces from the store and realises the stacking of delayed and weighted traces and their subsequent convolution with the `Source`'s STF. Finally, the `Engine` returns the synthetics for the request defined by the `Target` and `Source` objects. An example illustrating the the use of the *Pyrocko-GF* API is given in listing 1.

5    In practice, the described stacking of GF traces is computationally demanding. Stored GF traces are time dependent and differ in length. Consequently, a stacking of these GF traces requires time-shifting of GF traces and dynamic resizing of the output response. GF stores can become large, depending on resolution in time and space. To overcome limitations, we make use of virtual memory mapping of files. Parallelised C extensions manage queries to the database and efficiently stack the GF traces, according to the different component schemes. An adaptive stacking scheme is available for waveforms and static

10  models, which often have a very large number of source-receiver pairs. To ensure the correct complex interplay of functions software unit tests are carried out routinely.

The presented architecture in *Pyrocko-GF* enables simple implementation of custom point and extended dislocation sources, as well as STFs. The framework's flexibility allows modelling of finite earthquake ruptures with variable STFs across the rupture plane (Vasyura-Bathke et al., 2019). The complete documentation of *Pyrocko-GF*'s API is available online at https:

15  //pyrocko.org/docs/current/, featuring tutorials and examples of Python scripts and Jupyter notebooks.

**Listing 1.** Python script example for forward modelling a synthetic seismogram with the *Pyrocko-GF* framework. More complete examples are available on https://pyrocko.org/docs/current/library/examples/gf_forward.html

```python
1   from pyrocko import gf, io
2
3   engine = gf.LocalEngine(store_superdirs=['.'])
4   store_id = 'global_2s'
5
6   # Define three seismogram targets representing East-North-Vertical(up) channels
7   targets = [
8       gf.Target(
9           quantity='displacement',
10          lat=37.29, lon=-121.31,
11          store_id=store_id,
12          codes=('', 'STA', '', channel_code))
13      for channel_code in 'ENZ']
14
15  # Initialise a Double-Couple dislocation source
16  source_dc = gf.DCSource(lat=52.41, lon=13.06, magnitude=7.3)
17
18  # Create synthetic seismograms
19  response = engine.process(source_dc, targets)
20  synthetic_traces = response.pyrocko_traces()
21
22  # Save seismograms to Mini-SEED files
23  io.save(synthetic_traces, 'output/%(station)s-%(channel)s.mseed')
```





## 4   Applications

### 4.1   Seismic and acoustic source inversion

Challenges in seismic source inversions include computational aspects, as for instance the estimation of parameter uncertainties, and the aspect of a flexible definition of force representations for the specific rupture process under study. Several groups of
researchers use GF stores and the *Pyrocko* framework for source parameter estimation and centroid moment tensor inversion. Examples include nuclear explosion source studies (Cesca et al., 2017; Gäbler et al., 2018), anthropogenic triggered events (Sen et al., 2013; Ma et al., 2018; Grigoli et al., 2018), sinkhole processes (Dahm et al., 2011), local and regional tectonic earthquakes (e.g. Dahm et al., 2018), magmatic-induced low-frequency earthquakes (Hensch et al., 2019), earthquakes related to caldera collapses (Gudmundsson et al., 2016; Cesca et al., 2019), landslides (Kulikova et al., 2016), or meteor explosions
(Heimann et al., 2013). These examples show that *Pyrocko* GF stores (in some cases its predecessors) have been applied to synthesise data from global, regional and local seismic networks, with different frequency ranges and different wave modes (body waves, surface waves, W-phases, etc.). Also, the range of studied source types is wide including full moment tensors, double-couple, single-force sources, finite-extent earthquake sources and point or finite-extent volcanic sources.

A standard application is to estimate the parameters of a double-couple source from teleseismic data by using a GF store
based on a global velocity model. We provide a commented step-by-step code example for the real-data optimisation of the 2009 L'Aquila earthquake (Italy) in form of a Python *Jupyter notebook* at https://github.com/pyrocko/pyrocko-notebooks/blob/master/Waveform_Inversion-Double-Couple.ipynb.

A more exotic example is the source inversion of the Chelyabinsk Meteor's Terminal Explosion (Heimann et al., 2013). It demonstrates the flexibility towards specialised source-types and GF stores involving the atmospheric layers. The Chelyabinsk
Meteor's explosion occurred on 15 February 2013 near the town of Chelyabinsk, Russia, at an altitude of about 23 km. Through seismo-acoustic coupling, its shock-wave generated Rayleigh waves that were observed at epicentral distances of up to 4000 km. We analysed these recordings with a modelling/inversion approach. Synthetic GFs for atmospheric, surface, and underground sources were calculated using the QSSP program (Wang et al., 2017), which can handle the coupling of the solid Earth and atmosphere. The inversion for source location and strength, including error analysis, was performed using the
Kiwi Tools (Heimann, 2011), a software for earthquake source inversion. The GF store approach allowed us to cleanly separate the inversion from the GF generation task with only minimal modifications applied to the two distinct code bases. The GF store used for the Chelyabinsk Meteor explosion source inversion is available at https://greens-mill.pyrocko.org/chelyabinsk.

The separation of GF handling, source modelling and inversion/optimisation encourages designing modular software architectures with well-defined interfaces. At the time of writing, two feature-rich, open source earthquake analysis software
packages rely on *Pyrocko-GF* for modelling and GF handling: the Bayesian Earthquake Analysis Tool (BEAT, Vasyura-Bathke et al., 2019), which implements Bayesian inference for source characterisation, and the probabilistic earthquake source inversion framework *Grond* (Heimann et al., 2018), which implements a Bayesian bootstrap technique.



## 4.2 Inverting near-field surface displacements during magma chamber evacuation

In geophysics, static displacement in the near-field is often modelled assuming a homogeneous half-space, using analytical solutions (e.g. Mogi, 1958 or Okada, 1985). Many methods for modelling more realistic dislocation sources and media are available, but at higher computational cost. A GF store approach can compensate this obstacle and provide a convenient access

to a more complex modelling environment. Our *fomosto* tool facilitates the computation of static GF stores for layered media with its PSGRN-PSCMP (Wang et al., 2006) backend. To illustrate the use of *Pyrocko-GF* for static deformation modelling, let us examine a case study on the volcano-tectonic crisis in the Comoro Islands, which has started in 2018 offshore Mayotte Island (Cesca et al., 2019).

On Mayotte Island four GNSS stations showed large horizontal displacements and subsidence of up to 20 cm over a period

of more than eight months, the distinct horizontal displacements point towards a deflation source situated offshore. We used the few near-field GNSS data to complement the seismic analysis in constraining the location and character of the deflation source. From the seismic analysis they inferred an upward migration of the seismicity from  20 km to shallower crustal levels Cesca et al. (2019).

We tested different point dislocation sources and velocity/elasticity models against the observed GNSS data and evaluated

the different models in a Bayesian inversion, provided by the *Grond* framework (Heimann et al., 2018). We found that an isotropic source model in a homogeneous half-space cannot adequately represent the observations. However, the observations are compatible with a constrained moment tensor based point-source model, representative for a vertically elongated, deflating ellipsoid, similar to the point compound dislocation model (Nikkhoo et al., 2016) or an ellipsoidal cavity (Segall, 2010, Chap. 7). Such a tailored dislocation source and more realistic elasticity model derived from the AK135 model[1] (Kennett et al.,

1995) provide a more physical representation of the phenomenon and thus a better interpretation of the underlying geological processes.

In this case study, *Pyrocko-GF* provided an efficient modelling framework, where a freshly designed source model could be plugged in without requiring additional implementation work on the forward modelling itself. Seismic waveforms and static displacement are analysed in a homogeneous environment in terms of used Earth models and software toolset.

## 25 4.3 Ground motion and shake map simulation

GF stores can be used to explore attenuation functions of ground motions and their uncertainties that are related to the variability of the source and the structural parameters of the medium. Such explorations can be applied in scenario simulations to support risk assessment procedures. Furthermore, it is possible to generate near real-time ground motion forecasts after an earthquake. The *Pyrocko-GF* framework is sufficiently fast to allow for such near real-time waveform simulations. The advantage of

synthetic GF-based approaches to empirical relations of peak ground motion (Dahm et al., 2018) and magnitudes (Dahm et al., 2019) is; (1) that regional Earth structure models can be taken into account and (2) that earthquake sources can be simulated in regions where natural earthquakes have not been recorded. The latter situation also arises in most cases of induced seismicity,

---

[1]GF Store https://greens-mill.pyrocko.org/ak135_static-b18151

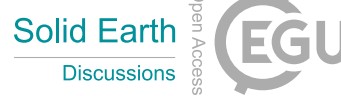



for which hazard and risk assessment is of particular importance. A challenge in induced and micro-earthquake ground motion simulations is, however, to realistically simulate synthetic seismograms at sufficiently high frequencies.

Dahm et al. (2018) used a synthetic GF approach to validate peak velocity attenuation relations in central Germany, where damaging earthquakes did strike in historical times, but not during the instrumental period. Two similar GF stores

5  were created to calculate peak ground velocity for two deep crustal earthquakes. One store contained GFs for a medium with a hard rock surface layer, while the other store had a soft sedimentary surface layer (find the corresponding GF stores *crust2_m5_hard_top_16Hz* and *crust2_m5_16Hz* at https://greens-mill.pyrocko.org. GFs are pre-calculated for distances between 0 and 300 km, for source depths ranging between 0 and 35 km, at intervals of 1 km with a sampling rate of 16 Hz. Peak ground velocities were extracted for 6 Hz low-pass filtered seismograms ($PGV_{6Hz}$). The $PGV_{6Hz}$ attenuation reproduced well

10  the observed averages and their variability for the two deep crustal earthquakes, while empirical attenuation relations derived for other regions over or under-predicted the observations. In Fig. 4 we show, as an example, $PGV_{6Hz}$ shake map simulations for a magnitude $M_W$ 6.3 earthquake scenario near the cities of Leipzig and Halle, Germany[2].

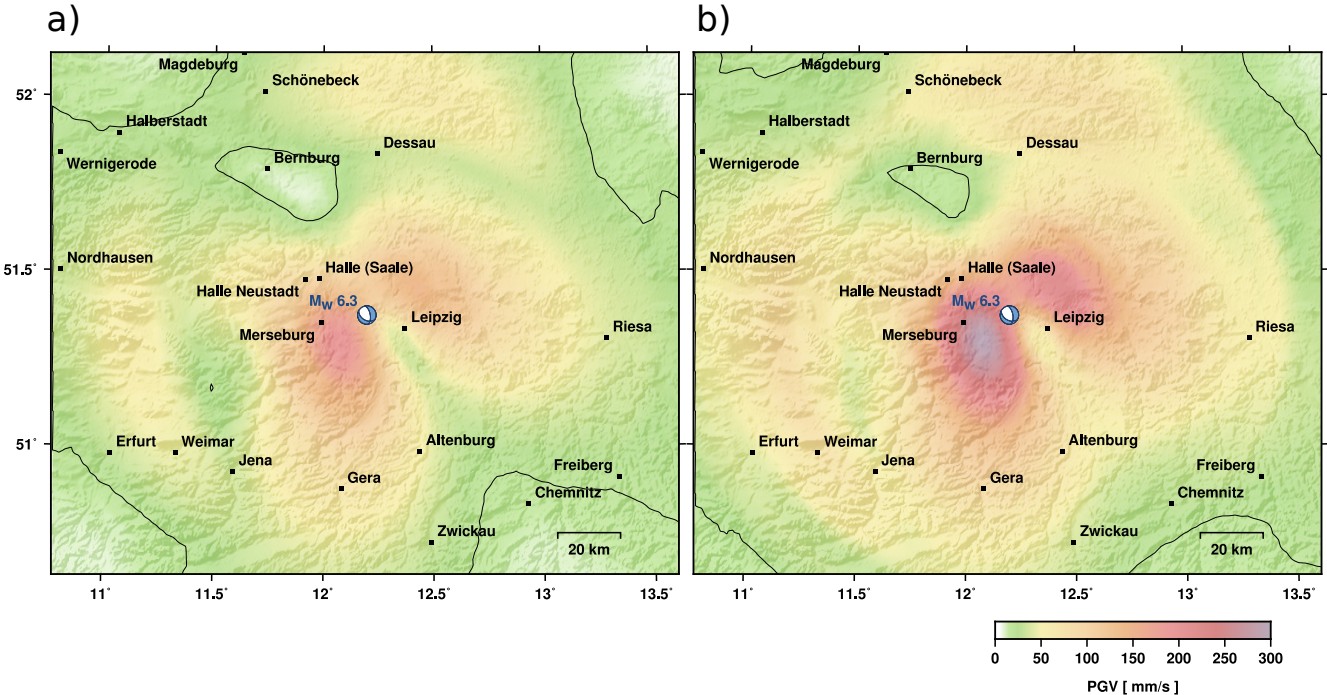

**Figure 4.** Shake map scenario of a $M_W$ 6.3 characteristic lower crust earthquake in 25 km depth between Halle and Leipzig, Germany. (a) Predicted $PGV_{6Hz}$ assuming hard rock upper layer model. (b) Predicted $PGV_{6Hz}$ assuming a 500 m thick soft layer beneath the surface with P and S wave velocities of 2.5 km/s and 1.2 km/s, respectively. The contour line shows the 20 mm/s threshold (see Dahm et al., 2018, for further explanation).

---

[2]See shake map code example at https://pyrocko.org/docs/current/examples/gf_forward.html



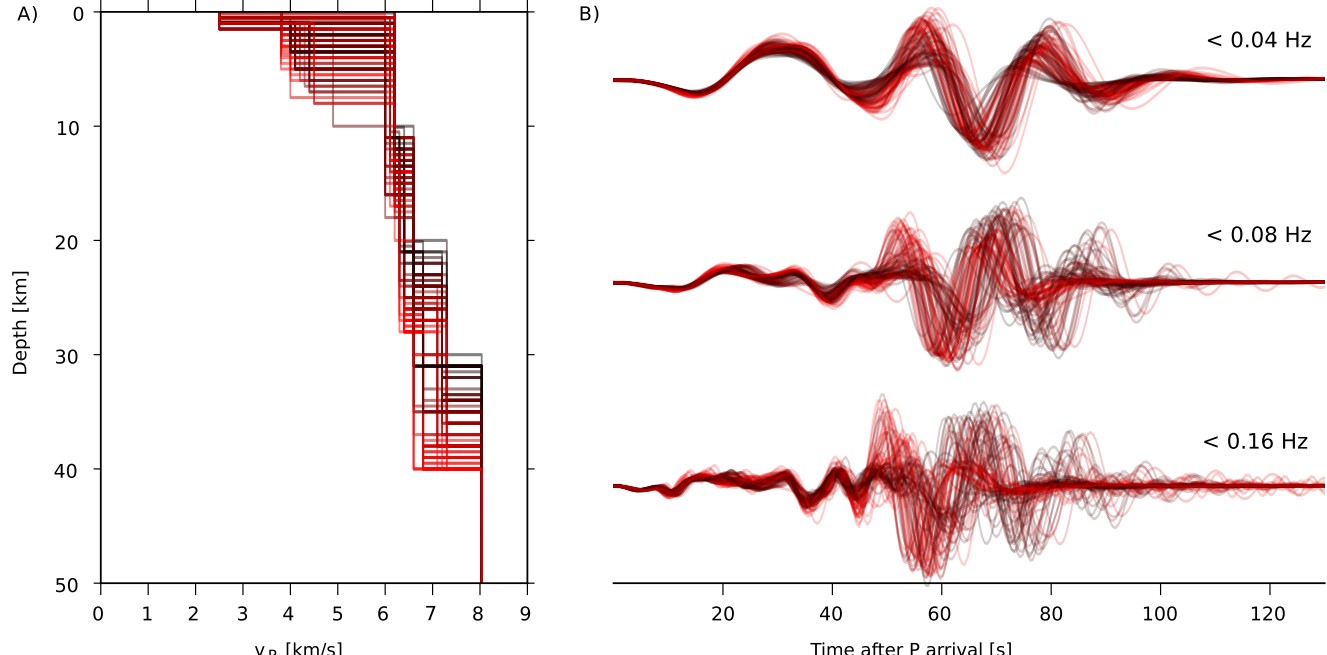

**Figure 5.** Illustration of waveform sensitivities caused by crustal velocity variations. (a) Colour-coded crustal P-wave velocity profiles as a function of depth. Darker colours correspond to lower crustal thickness. (b) Synthetic, low-pass filtered seismograms of a double-couple point source corresponding to the P-wave velocity structure in (a) at 278 km distance).

Peak ground motions of displacements or velocities are used to estimate earthquake magnitudes. Usually, empirical attenuation relations are used to normalise measured peak values and to estimate station magnitudes. The empirical relations are often poorly calibrated for the region under study, and only valid for a narrow range of instruments, source depths and strengths. GF stores can overcome some of the limitations of empirical attenuation curves in magnitude scales and can be used to estimate
5   moment magnitudes from peak ground motions. Examples are provided in Dahm et al. (2019) for local magnitude estimations. Figure 5 gives an example how the effect of different crustal models on full waveforms can be simulated on-the-fly[3]. This can be used, for instance, to explore the station variability of global surface wave magnitude scales.

---

[3]See Rayleigh wave waveform variability code example at https://pyrocko.org/docs/current/examples/gf_forward.html



## 4.4 Array and network design optimisation by full waveform simulation

The simulation of full waveforms for the directed Monte-Carlo optimization of seismic networks is computationally demanding and not standard in seismology. López-Comino et al. (2017) used synthetic GF stores for the simulation of full waveforms to predict the network performance and spatial and temporal variation of the magnitude of completeness before the deployment of
stations. Synthetic velocity seismograms were generated and modified by real recorded noise based on a randomised catalogue of synthetic earthquake sources for a given source volume in the area under investigation. The seismograms were then used in a full waveform event detector to evaluate the expected magnitude of completeness.

For the design of a small-aperture array for the study of local micro-earthquakes, a synthetic GF store has been used by Karamzadeh Toularoud et al. (2019). For a given distribution of synthetic hypocenters and source mechanisms that simulate
an earthquake swarm, the authors randomly generated array locations and station configurations, for which synthetic seismic waveforms were computed. For these synthetic setups the expected beam power and the slowness calculation error was optimised, which allowed the authors to judge on a preferred array installation configuration. The array setup was fixed to use seven stations with three components. 4000 different station configurations have been tested with 100 different sources each. Hence, a total number of 8.4 million traces had to be generated for the analysis. A desktop computer was sufficient for the com-
putational task. Karamzadeh Toularoud et al. (2019) used the GF store https://greens-mill.pyrocko.org/vogtland_malek2004_ sed_HR-95ef03.

## 5 Conclusions

*Pyrocko-GF* is a Python module that provides functionality to calculate Green's functions (GFs) and to organise these in a store. In *Pyrocko-GF*, backends utilise different numerical calculation methods for GFs, making the stores independent of the
actual type of GFs. The command-line interface tool *fomosto* within the *Pyrocko-GF* framework manages the generation of GFs through the backends. *fomosto* also facilitates the comparison, visualisation and quality check of stored GFs. The software is open source and encourages the contribution of use-specific extensions.

Our implementations support computation of GFs useful for earthquake-related problems, which include GFs for broadband global, regional and local seismograms, optionally with near-field terms, as well as static surface displacements as measured by
GNSS sensors and/or through InSAR. Furthermore *Pyrocko-GF* stores are prepared to store infrasound, oceanic infragravity, poro-elastic GFs and elasto-gravitational GFs. On the GF online platform *Green's Mill* (https://greens-mill.pyrocko.org) we provide GF stores for download that are useful for many standard earthquake-related applications using global velocity models.

The *Pyrocko-GF* framework allows to efficiently access GF stores to synthesise source-specific responses inside the medium for comparison with different observations. Sources as well as source-time-functions can easily be customised without limita-
tions on the source complexity. We discussed seismological applications using *Pyrocko-GF* stores to simulate seismic waveforms, static surface displacements and peak ground motion caused by earthquakes, volcanic activity and exploding meteoroids. The range of applications is not limited to generalised force couples or single forces. With new backends also flow rate, pres-

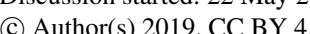
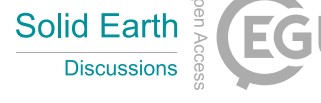

sure pulse or any other type of excitation and observable, which are representable using the GF approach, could be included in a *Pyrocko-GF* store.

*Code and data availability.* The presented software is available as open source under the GNU General Public License v3 and can be downloaded from https://github.com/pyrocko. The documentation and examples are available at https://pyrocko.org. Pre-computed Green's

function stores can be browsed and downloaded at https://greens-mill.pyrocko.org/.

*Author contributions.* S.H. and T.D. are the scientific Pyrocko project leader, supported by H.S. and H.V.-B. as scientific project partners. S.H. leads the software implementation for which H.V.-B., M.I. and M.K have contributed significantly and in many ways, e. g. by writing parts of the codes. All authors have contributed in the software design, the code testing and the manuscript writing.

*Acknowledgements.* H.S., A.S. and M.I. acknowledge funding by the German Research Foundation DFG through an Emmy-Noether Young-

Researcher-Grant (276464525).

H.V.-B. was financially supported by King Abdullah University of Science and Technology (KAUST) BAS/1/1353-01-01 as well as by Geo.X, the Research Network for Geosciences in Berlin and Potsdam (Project number: SO_087_GeoX). T.D. and S.H. acknowledge co-funding by SERA (EU Grant 730900). Timothy Willey helped to implement fomosto report plots. University of Rheno, Nevada for processing and providing GNSS deformation data for Mayotte Island (Blewitt et al., 2018).

*Competing interests.* The authors declare that they have no conflict of interest.

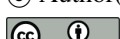



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
