# Peer review of "A Python framework for efficient use of pre-computed Green's functions in seismological and other physical forward and inverse source problems"

_Solid Earth, 2019_

## Referee Comment (RC1) · Tim Greenfield (Referee) · 20 Jun 2019

Heimann et al. present a utility to generate, store and utilise Green's Functions (GF) for a variety of geophysical applications. The manuscript simply and concisely details the mathematics underpinning the creation of the GFs then explain how the GFs are combined to create synthetic seismograms within the fomosto user interface. Finally, a number of potential and realised applications are described. The ability to easily create and access GFs is a useful tool in the geophysicists toolbox and this software package is a welcome addition to tackling geophysical problems in a python environment.

The text is succinct and well-presented, containing enough detail for a potential enduser to begin using the package without drowning them in unnecessary complications. I have no issues with the content or the structure generally and struggled to find any grammatical or typographical errors (any I found are detailed below). I recommend the text be accepted after correction of the (few) technical corrections.

Tim Greenfield

Technical corrections

Page 2, Line 4: The sentence starting "In physics..." is awkward. I suggest changing it to "Physically..."

Page 3, Line 1: the second "let" in the sentence is not required.

Page 3, Line 2: I would move the "then" to between "is" and "given"

Page 12, Line 9: The sentence starting "Yet..." doesn't sound right. I suggest changing it to "In addition..."

---

## Editor Comment (EC1) · Caroline Beghein (Editor) · 17 Jul 2019

Dear authors,

We are still waiting for a second review of your paper. In the meantime, I received some comments from Dr. Martin van Driel. He was unable to write a full, detailed review, but I wanted to share the comments he sent me. See below:

From M. van Driel:

- There is a paper about IRIS's syngine service, not just a website, so it should be referenced: Krischer, L., Hutko, A., van Driel, M., Stähler, S., Bahavar, M., Trabant, C.,

and Nissen-Meyer, T. (2017). On-demand custom broadband synthetic seismo-grams, Seismol. Res. Lett. 88, no. 4, https://doi.org/10.1785/0220160210.

- P8, L13: 'The program INSTASEIS (Van Driel et al., 2015) is suited to calculate ap-proximative wave solutions for a 3D Earth structure with radial symmetry' > This state-ment is completely wrong, instaseis (not fortran, hence no capitalization in the name), does not solve the wave equation and is hence also not limited to radial symmetry. We use AxiSEM to compute the databases, which is where the limitation to spherical sym-metry comes from, but that is completely independent of instaseis, which could easily include any other database source.

- One of my main concerns when I wrote instaseis was to be accurate enough in space to place receivers anywhere and still get the phase correct so you can apply array methods. Similarly, we wanted to be accurate in the presence of discontinuities. This is in fact a non-trivial interpolation, and we approach it using the spectral element basis which we discus in some length in our paper. The statement on P10 L2 makes me very suspicious that this issue is treated appropriately here.

---

## Referee Comment (RC2) · Anonymous Referee #2 · 18 Jul 2019

This is a well-written technical paper. The code and the web service look well-established for user applications. But two points are not clear to me.

1. model assumption, 1D/2D/3D?

In Section 2.2, the authors present the theory for layered models, for which we can take advantage of the four radiation patterns of the wave solution, i.e., eq.8, the same as those in Nissen-Meyer et al. 2007.

First, the correctness of Section 2.2 does not necessarily require the model to be layered; it only requires the model to be axisymmetric with respect to the source, that is, $Vs(r, theta, phi) = Vs(r, theta)$, sometimes referred to as 2D in-plane models. A layered

model is a stronger assumption: Vs(r, theta, phi) = Vs(r), or a 1D model.

It is not clear if their code is designed only for 1D layered models or it can deal with 2D in-plane models or even an arbitrary 3D model. The authors mentioned Crust 2.0, which is inherently a 3D model. On their website, I see they have computed GF's for 1D profiles of Crust 2.0.

Does the code support both spherical and Cartesian geometry (for local)? Pressure source in a fluid ocean? ...

The model assumptions and requirements for the code should be made explicit in the abstract and conclusion.

2. comparison with Instaseis/Syngine

About Instaseis/Syngine, the authors wrote

"However, these rigid database schemes are restricted to the modelling method that has been used to create them, and they are confined to specific moment tensor applications."

"specific moment tensor" may not be correct. As a user of Instaseis/Syngine, so far as I know, Instaseis and Syngine accept an arbitrary moment tensor (based on eq.8) and Instaseis can also handle point forces (for receiver-wise reciprocity database).

Because their contribution has a similar purpose and follows similar principles to Instaseis/Syngine, its advantage and generalisation should be correctly and clearly explained. Two advantages seem clear to me: an Instaseis database can be generated only by AxiSEM, while a Pyrocko database is compatible with any forward simulation methods, as claimed by the authors; besides, Instaseis/Syngine is only for global scale.

---

## Author Comment (AC1) · 28 Aug 2019

Dear Dr. Greenfield,

Thank you very much for reviewing our manuscript. In the following paragraphs we reply each of your comments. For each reply we detail what we changed in the manuscript accordingly.

Kind regards,

Sebastian Heimann

**I.**

**Comment:** Page 2, Line 4: The sentence starting "In physics..." is awkward. I suggest changing into "Physically..."

Thank you for pointing this out. We decided to go for "In the field of physics, ..." to make the meaning clearer without changing it.

**Changed:** "In the field of physics, ..."

**II.**

**Comment:** Page 3, Line 1: the second "let" in the sentence is not required.

Done.

**Changed:** "... and  f be the force vector ..."

**III.**

**Comment:** Page 3, Line 2: I would move the "then" to between "is" and "given"

Done.

**Changed:** "...  the GF representation is then given by ..."

**IV.**

**Comment:** Page 12, Line 9: The sentence starting "Yet..." doesn't sound right. I suggest changing it to "In addition..."

Done.

**Changed:** "In addition, seismic targets can provide derivatives of the synthetic displacements ..."

---

## Author Comment (AC2) · 28 Aug 2019

Dear Dr. van Driel,

Thank you very much for reviewing our manuscript. In the following paragraphs we reply each of your comments. For each reply we detail what we changed in the manuscript accordingly.

Kind regards,

Sebastian Heimann

**Comments by Martin Van Driel**

**I.**

**Comment:** There is a paper about IRIS's syngine service, not just a website, so it should be referenced: Krischer, L., Hutko, A., van Driel, M., Stähler, S., Bahavar, M., Trabant, and Nissen-Meyer, T. (2017). On-demand custom broadband synthetic seismograms, Seismol. Res. Lett. 88, no. 4, https://doi.org/10.1785/0220160210.

Thank you. We added this missing reference.

**Changed:** "... for seismological applications, like e. g. Instaseis (Van Driel et al., 2015) or Syngine (IRIS DMC, 2015; Krischer et al., 2017)."

**II.**

**Comment:** P8, L13: "The program INSTASEIS (Van Driel et al., 2015) is suited to calculate approximative wave solutions for a 3D Earth structure with radial symmetry" - This statement is completely wrong, instaseis (not fortran, hence no capitalization in the name), does not solve the wave equation and is hence also not limited to radial symmetry. We use AxiSEM to compute the databases, which is where the limitation to spherical symmetry comes from, but that is completely independent of instaseis, which could easily include any other database source

We apologize for this incorrect statement. We changed the description accordingly.

**Changed:** "   The program AxiSEM (Nissen-Meyer et al., 2014), as employed by Instaseis (Van Driel et al., 2015), provides seismic wavefields for a 3D axisymmetric Earth structure."

**III.**

**Comment:** One of my main concerns when I wrote instaseis was to be accurate enough in space to place receivers anywhere and still get the phase correct so you can apply array methods. Similarly, we wanted to be accurate in the presence of discontinuities. This is in fact a non-trivial interpolation, and we approach it using the spectral element basis, which we discus in some length in our paper. The statement on P10 L2 makes me very suspicious that this issue is treated appropriately here.

Thank you for this comment. We are aware of the importance of interpolation effects and the need for appropriately fine GF component grid spacing for sufficient accuracy in space and time. We agree that the reader should be made aware of it more explicitly and provide advise. We added text to that effect in Section 3.3 "Source Design" and explain the influence of the user defined grid spacing and interpolation with an expanded paragraph and an additional figure.

**Changed:**

Section 3.3, from second paragraph on: "The calculation of an observed quantity of interest (e. g. seismic waveforms) for a point source with delta-force excitation at a particular source-receiver constellation is given in section 2. As described above, the specific combination of GF components is defined by the source type and the observed type of quantity (e. g. a full moment tensor or a single force, generating far-field waveforms or surface displacements, see Tab. 1).

The observed quantity at the receiver is a linearly weighted combination of the spatially closest GF components. Often, this requires interpolation to match the requested source-receiver configuration. The interpolation   of GF components requires an ap-

[Figure]

Figure 1: Comparison of interpolation artifacts on synthetic waveforms with different GF grid spacings (a: nearest-neighbour, b: multi-linear). Shown are vertical component displacements, based on GF stores with 1 km, 4 km and 8 km spatial grid spacing (blue, green and orange, respectively) against a reference for the exact source-receiver distance (QSEIS; red). The sampling rate is 2 Hz and the signal contains information up to close to the Nyquist frequency (1 Hz). The waveforms are filtered with a pass band from 0.05 Hz to 0.1 Hz, after interpolation. The medium is layered with important discontinuities of upper crust, lower crust and mantle at 20 km and 35 km depth, respectively. The slowest seismic velocity in the medium is 3.5 km/s. The waveform is simulated for a 10 km deep moment tensor source at a distance of 553.3 km.

propriate density of grid nodes to result in seismograms that are accurate in amplitude and phase (Fig. 1). Furthermore, simulated seismograms vary for different interpolation methods such as nearest-neighbour or multi-linear  interpolation. For standard applications with multi-linear interpolation, we could require that the grid spacing $d_{grid}$ should be less than a quarter of the  minimum wavelength. It can be estimated using the minimum wave velocity $v_{min}$ and the maximum signal frequency of the GF traces $f_{max}$ with $d_{grid} = v_{min}/(4 f_{max})$. For applications requiring higher accuracy, a smaller grid spacing must be used. For static displacements in the near-field of finite sources discussed below, an appropriate gird spacing is smaller than half the minimum source-receiver distance. In general, smaller grid spacing leads to higher accuracy at the cost of forward-modelling performance and larger GF stores. Interpolation of GF components in the spectral domain is superior but computationally more demanding (Gülünay, 2003)."

---

## Author Comment (AC3) · 28 Aug 2019

Dear Anonymous Referee #2,

Thank you very much for reviewing our manuscript. In the following paragraphs we reply each of your comments. For each reply we detail what we changed in the manuscript accordingly.

Kind regards,

Sebastian Heimann

**Your review**

**Comment:** This is a well-written technical paper. The code and the web service look well-established for user applications. But two points are not clear to me.

**I.**

**Comment:** Model assumption, 1D/2D/3D? In Section 2.2, the authors present the theory for layered models, for which we can take advantage of the four radiation patterns of the wave solution, i.e., eq.8, the same as those in Nissen-Meyer et al. 2007. First, the correctness of Section 2.2 does not necessarily require the model to be layered; it only requires the model to be axisymmetric with respect to the source, that is, $V_S(r, \theta, \phi) = V_S(r, \theta)$, sometimes referred to as 2D in-plane models. A layered model is a stronger assumption: $V_S(r, \theta, \phi) = V_S(r)$, or a 1D model. It is not clear if their code is designed only for 1D layered models or it can deal with 2D in-plane models or even an arbitrary 3D model. The authors mentioned Crust 2.0, which is inherently a 3D model. On their website, I see they have computed GF's for 1D profiles of Crust 2.0.

Thank you for pointing out the need for clarifications.

Pyrocko-GF can handle both: the axisymmetric special case (store types A and B) and a case suitable for 3D heterogeneous media based on the concept of a (densely) gridded source volume and fixed receiver positions (store type C).

On the issue with axisymmetric vs. layered media: While we clearly state in Section 2 that the functions build up on cylindrical symmetry (P6, L13 and L23), we put a focus on layered media due to their more versatile applicability. Nevertheless, the architecture of Pyrocko-GF does not hinder the user to create GF stores for axisymmetric media. Only care has to be taken that such a GF store can only be valid for a fixed epicenter (or fixed receiver location), limiting its use to special problem geometries (e.g. sources below the center of a volcano).

The computation backends currently supported by the Pyrocko-GF's fomosto tool, QSEIS, QSSP, PSGRN, and PSCMP only handle layered velocity models at the moment. Therefore, at the moment, custom import of the GF traces is required for GFs from axisymmetric or 3D media modellers.

**Changes**

We have adjusted the text in several places to make these issues more clear to the reader.

Section 2.2. Corrected statement: "The number of independent GF components can be further reduced for  media of cylindrical symmetry..."'

Section 3.1. Second paragraph we specify in the second and third sentences: "The physics of the modelled process  and the symmetry of the medium determine the number of GF components ...", and "For example, in a medium of cylindrical symmetry, with a planar layered medium , and at short distances, the simulation of a full moment tensor requires 10 GF components, ...", respectively.

Section 3.3. Second paragraph, we added: As described above, the specific combination of GF components is defined by the source type and the observed type of quantity (e.g. full moment tensor or a single force, generating far field waveforms or surface displacements, see Tab. 1)..

In the caption of Tab. 1 we clarify: "Configuration type (C) is used for three-dimensionally heterogeneous media with sources in a  defined source volume and receivers at fixed positions."

**II.**

**Comment:** Does the code support both spherical and Cartesian geometry (for local)?

Yes, in our framework (Pyrocko) we use for the definition of location coordinates that are generally given with five numbers. These are geographical coordinates and depths as well as a local Cartesian offset in horizontal east and north direction. For example, a group of locations (stations or sources) can have the same latitude and longitude values but different east shifts, north shifts and depths. This is useful for small scale networks and/or seismicity. Locations across the globe may have latitudes and longitudes with zero Cartesian offsets in east and north. The framework offers many functions to derive distances and azimuths on a sphere and distances are generally distances on spherical surfaces. The coordinate definition and the functions ease the use of GF calculation methods that require either geographical coordinates for calculations with spherical models (e.g. QSSP) or Cartesian input for flat-earth methods (e.g. mode in QSEIS). The GF architecture is in general independent of the coordinate systems. The stores are sorted grids of functions, which the user configures. To make this clearer we complemented the text with some changes and additions.

**Changes**

In the third paragraph of Section 3.1 after the first sentence we added: "The 'config' file also contains the medium model definitions, i.e. seismic velocities, densities, attenuation coefficients etc. Through selection of a component scheme and configuration type, it defines the mapping used to transform physical source/receiver coordinates into the file lookup indices used in the index file."

In Section 3.5 the first paragraph is expanded with: "... `Source` and `Target` locations are specified as geographic coordinates with an optional Cartesian offset. This design allows the user to handle global, local, as well as mixed setups. These different setups are achieved, respectively, by either setting the Cartesian offset to zero, by setting the geographic coordinates to a common reference location, or by using a combination of both."

**III.**

**Comment:** Pressure source in a fluid ocean? ... The model assumptions and requirements for the code should be made explicit in the abstract and conclusions.

We are not sure if we understand this point entirely, because we nowhere mention a fluid ocean. If you are asking if Pyrocko-GF provides a forward modelling scheme for a pressure source in a fluid ocean, yes, it does. A fluid ocean has a P-wave velocity changing with depth, but zero S-wave velocities. This can be defined in the 'config' file of a GF store. Using further a component scheme 'elastic2' for an isotropic moment tensor and a store type B would do the trick for the forward calculation of P-Waves at different depth. The store could be filled with GFs computed using the QSEIS or QSSP backend. We introduce the concept and abilities in Sections 3.1 and 3.2. The model assumptions are therefore defined mainly by the GF calculation method, the backend, employed. Otherwise, we point out in the paper that Pyrocko-GF is explicitly open for many different medium and source models with different assumptions and requirements. In the paper we introduce the conceptual architecture for storage. This is why we do not limit the applicability to specific model assumptions and requirements up front so much. However, with this comment we see the need for some more clarifications. Specifically, to clarify the potential use of Pyrocko-GF and the model assumptions for the ready-to-use software solutions to which we refer to as "the Pyrocko framework".

**Changes**

In the abstract, line 10, we changed: "The framework  aids in the creation of such GF stores by interfacing a suite of established numerical forward-modelling codes in seismology  (computational backends). So far interfaces to backends for layered Earth model cases are provided, however, the architecture of Pyrocko-GF is designed to cover backends for other geometries (e.g. full 3D heterogeneous media) and other physical quantities (e.g. gravity, pressure, tilt). Therefore, Pyrocko-GF defines an extensible GF storage format..."

In Section 3.2, we added more information about the model assumptions of the GF calculation methods: "These GF calculation methods assume horizontally layered medium models, apart from AxiSEM, which requires axisymmetric heterogeneity only."

In the conclusions, we changed: "The command-line interface tool *fomosto* within the Pyrocko-GF framework manages the generation of GFs through the  computational backends, which so far exist for layered media. *fomosto* also facilitates the comparison, visualisation and quality check of stored GFs. The software is open source and encourages the contribution of use-specific extensions and adding interfaces to other computational backends."

**IV.**

**Comment:** comparison with Instaseis/SyngineAbout Instaseis/Syngine, the authors wrote "However, these rigid database schemes are restricted to the modelling method that has been used to create them, and they are confined to specific moment tensor applications." "specific moment tensor" may not be correct. As a user of Instaseis/Syngine, so far as I know, Instaseis and Syngine accept an arbitrary moment tensor (based on eq.8) and Instaseis can also handle point forces (for receiver-wise reciprocity database).

Thank you for that comment and we apologize for that incorrect statement. Indeed instaseis/syngine can simulate other sources as well.

**Changes**

Paragraph 4. Introduction, last sentence, we corrected "However,  rigid database schemes  can be restricted to the modelling method  which has been used to create them."

**V.**

**Comment:** Because their contribution has a similar purpose and follows similar principles to Instaseis/Syngine, its advantage and generalisation should be correctly and clearly explained. Two advantages seem clear to me: an Instaseis database can be generated only by AxiSEM, while a Pyrocko database is compatible with any forward simulation methods, as claimed by the authors; besides, Instaseis/Syngine is only for global scale.

We have corrected a few unfortunate misrepresentations of Instaseis. It generally can support other GF calculation methods than AxiSEM in the future, as stated in the comment by Van Driel. Also, Instaseis can be applied regionally. We see the Pyrocko-GF software and the Pyrocko framework as a useful complementary code to Instaseis/Syngine. We support other GF calculation methods. We offer the same and some more source types compared to Instaseis, e.g. a direct support four building finite sources, but this may evolve fast in the near future. Whichever code is better suited depends on the problems the user wants to tackle. A detailed comparison is beyond the scope of the presentation and could be outdated soon. In this regard, we added some numbers on the performance to give the user a better feeling on what computational costs may be expected for some setups.

**Changes**

Section 3.4. Added paragraph: "The computational effort to create a GF store depends on the complexity of the medium model, the temporal and spatial sampling and the duration of the desired waveforms. For example, a global database based on the PREM model, calculated with QSSP, with 2 s sampling and 4 km spatial spacing in distance and source depth, requires an effort of 19 h on a 100-core Intel Xeon E7-8890 high-performance computer and uses 52 GB of disk space. For comparison, regional GF stores at 2 Hz maximum frequency are built within hours on modern desktop computers.